# Towards an Adversarially Robust Normalization Approach

## Abstract

Batch Normalization (BatchNorm) has shown to be effective for improving and accelerating the training of deep neural networks. However, recently it has been shown that it is also vulnerable to adversarial perturbations. In this work, we aim to investigate the cause of adversarial vulnerability of the BatchNorm. We hypothesize that the use of different normalization statistics during training and inference (mini-batch statistics for training and moving average of these values at inference) is the main cause of this adversarial vulnerability in the BatchNorm layer. We empirically proved this by experiments on various neural network architectures and datasets. Furthermore, we introduce Robust Normalization (RobustNorm) and experimentally show that it is not only resilient to adversarial perturbation but also inherit the benefits of BatchNorm.

## 1 Introduction

In spite of their impressive performance on challenging tasks in computer vision such as image classification and semantic segmentation, deep neural networks (DNNs) are shown to be highly vulnerable to adversarial examples, i.e. carefully crafted samples which look similar to natural images but designed to mislead a trained neural network model (Goodfellow et al., 2014; Nguyen et al., 2015; Carlini & Wagner, 2017). Designing defense mechanisms against these adversarial perturbations has been subjected to much research recently (Xie et al., 2019; Madry et al., 2017; Tramèr et al., 2017; Papernot et al., 2016).

Meanwhile, Batch Normalization (BatchNorm or BN) (Ioffe & Szegedy, 2015) has successfully proliferated throughout all areas of deep learning as it enables stable training, higher learning rates, faster convergence, and higher generalization accuracy. Initially, the effectiveness of the BatchNorm has been attributed to its ability to eliminate the *internal covariate shift* (ICS), the tendency of the distribution of activations to drift during training. However, later on, alternative reasons including avoiding exploding activations, smooth loss landscape, reducing the sensitivity to initialization, etc. have also been proposed as the basis of BatchNorm's success (Santurkar et al., 2018; Bjorck et al., 2018; Luo et al., 2018).

While there exist a plethora of reasons for the adversarial vulnerability of deep neural networks (Jacobsen et al., 2018; Simon-Gabriel et al., 2018), a recent study by Galloway et al. (2019) showed that BatchNorm is one of the reasons for this vulnerability. Specifically, they empirically showed that removing the BatchNorm layer enhances robustness against adversarial perturbations. However, removal of BatchNorm also means a sacrifice of benefits such as the use of higher learning rates, faster convergence, and significant improvement in the clean test set accuracy among many others.

In this paper, we propose a new perspective regarding the adversarial vulnerability of the BatchNorm layer. Specifically, we probe why BatchNorm layer causes the adversarial vulnerability. We hypothesize that the use of different normalization statistics during training and inference phase (mini-batch statistics for training and moving average of these statistics also called tracking, at inference time) is the cause of this adversarial

vulnerability of the BatchNorm layer. Our experiments show that by removing this part of the BatchNorm, the robustness of the network increases by 20%. Similarly, robustness can further be enhanced by up to 30% after adversarial training. However, by removing the tracking part, the test accuracy on the clean images drops significantly ( though better than without normalization). To circumvent this issue, we propose Robust Normalization (RobustNorm or RN). Our experiments demonstrate that *RobustNorm* not only significantly improve the test performance of adversarially-trained DNNs but is also able to achieve the comparable test accuracy to that of BatchNorm on unperturbed datasets. We perform numerical experiments over standard datasets and DNN architectures. In almost all of our experiments, we obtain a better adversarial robustness performance on perturbed examples for training with natural as well as adversarial training.

## 2 BACKGROUND

We consider a standard classification task for data, having underlying distribution denoted as $\mathcal{D}$, over the pair of examples $\boldsymbol{x} \in \mathbb{R}^n$ and corresponding true labels $y \in \{1, 2, ..., k\}$ where $k$ represents different labels. We denote deep neural network (DNN) as a function $\mathcal{F}_{\boldsymbol{\theta}}(\boldsymbol{x})$, where $\boldsymbol{\theta}$ denotes trainable parameters. $\boldsymbol{\theta}$ is learned by minimizing a loss function $\mathcal{L}(\boldsymbol{x}, y)$ with training data $\boldsymbol{x}, y$. The output of the DNN is a feature representation $\boldsymbol{f} \in \mathbb{R}^d$, that we give input to a classifier $\mathcal{C} : \mathbb{R}^n \rightarrow \{1, 2, ..., k\}$. The objective of the adversary is to add the additive perturbation $\delta \in \mathbb{R}^n$ under the constrain that the generated adversarial sample $\boldsymbol{x}_{adv} = \boldsymbol{x} + \delta$ that looks visually similar to the true image $\boldsymbol{x}$, and for which the corresponding labels are not same i.e. $\mathcal{C}(\boldsymbol{x}) \neq \mathcal{C}(\boldsymbol{x}_{adv})$. In this work, we have added the perturbation via following well-known adversarial attack approaches.

**Fast Gradient Sign Method:** Given an input image $\boldsymbol{x}$ along with its corresponding true label $y$, FGSM Goodfellow et al. (2014) aims to generate the adversarial image $\boldsymbol{x}_{adv}$ as,

$$\boldsymbol{x}_{adv} = \boldsymbol{x} + \epsilon \cdot \text{sign}(\nabla_{\boldsymbol{x}} \mathcal{L}(\boldsymbol{x}, y), \tag{1}$$

where $\epsilon$ is the perturbation budget that is chosen to be sufficiently small. We use two of its variants: Gradient (Grad) where graidents are used and Gradient sign (GradSign) which is similar to 1.

**Basic Iterative Method (BIM):** BIM (Kurakin et al., 2016) is a straight forward extension of FGSM, that applies it multiple times with a smaller step size. Specifically,

$$\boldsymbol{x}_{adv}^0 = \boldsymbol{x}, \quad \boldsymbol{x}_{adv}^N = \text{Clip}_{\boldsymbol{x}, \epsilon} \{ \boldsymbol{x}_{adv}^{N-1} + \alpha \cdot \text{sign}(\nabla_{\boldsymbol{x}} \mathcal{L}(\boldsymbol{x}_{adv}^{N-1}, y)) \} \tag{2}$$

where $\boldsymbol{x}_{adv}^0$ is the clean image and $N$ denotes iteration number.

**Carlini-Wagner attack (CW):** CW is an effective optimization-based attack model introduced by Carlini & Wagner (2017). It works by definining an auxlary variable $\vartheta$ and minimizes the following objective functions

$$\min_{\vartheta} \| \frac{1}{2}(\tanh(\vartheta) + 1) - \boldsymbol{x} \| + c.f(\frac{1}{2}(\tanh(\vartheta) + 1)), \tag{3}$$

where $\frac{1}{2}(\tanh(\vartheta) + 1) - \boldsymbol{x}$ is the perturbation $\delta$, c is a scalar constant, and f(.) is defined as:

$$f(\boldsymbol{x}_{\text{adv}}) = \max(\mathcal{Z}(\boldsymbol{x}_{adv})_{\boldsymbol{y}} - \max\{\mathcal{Z}(\boldsymbol{x}_{adv})_k : k \neq \boldsymbol{y}\}, -\varrho)). \tag{4}$$

Here, $\varrho$ is to control the adversarial sample's confidence and $\mathcal{Z}_{\boldsymbol{x}_{adv}}$ are the logits values for class $k$.

**Projected Gradient Descent (PGD):** PGD perturbs the true image $\boldsymbol{x}$ for total number of $N$ steps with smaller step sizes. After each step of perturbation, PGD projects the adversarial example back onto the $\epsilon$-ball of normal image $\boldsymbol{x}$ , if it goes beyond the $\epsilon$-ball. Specifically,

$$\boldsymbol{x}_{adv}^N = \Pi(\boldsymbol{x}_{adv}^{N-1} + \alpha.\text{sign}(\nabla_{\boldsymbol{x}} \mathcal{L}(\boldsymbol{x}_{adv}^{N-1}, y))), \tag{5}$$

where $\Pi$ is the projection operator, $\alpha$ is step size, and $\boldsymbol{x}^N$ denotes adversarial example at the $N$-th step. We have used $\ell_\infty$ norm as a distance measure. **Gaussian Noise** For comparison purposes, we also have used Gaussian noise with 0 mean and 0.25 variance.

### 2.0.1 ADVERSARIAL TRAINING:

It has been shown that empirical risk minimization using only clean images for training can decrease the robustness performance of DNNs. A standard approach to achieve the adversarial robustness in DNNs is adversarial training which involves fitting a classifier $\mathcal{C}$ on adversarially-perturbed samples along with clean images. We have used PGD based adversarial training which has shown to be effective against many first-order adversaries (Madry et al., 2017) unlike other methods which overfit for on a single attack.

### 2.1 EXPERIMENTAL SETUP

We have used two network architectures, Resnet He et al. (2016) with 20,38 and 50 layers and VGG Simonyan & Zisserman (2014) with 11 and 16 layers. We have used CIFAR10 and 100 datasets (Krizhevsky et al., 2009) for all the evaluations. We have used term natural training for training with clean images while adversarial training is done with PGD based method formulated by Madry et al. (2017). We have always used a learning rate of 0.1 except for no normalization scenarios where convergence is not possible with higher learning rates. In such cases, we have used a learning rate of 0.001. We decrease the learning rate 10 times after 120 epochs and trained all the networks for 164 epochs. For robustness evaluations, we have used $\epsilon$=0.03/1 for most of the experiments and used 20 epochs for all the iterative attacks. We also have tested the model on different noise levels ranging from 0.003/1 to 0.9/1.

## 3 HOW DOES BATCHNORM CAUSES ADVERSARIAL VULNERABILITY

### 3.1 HOW BATCHNORM WORKS

In this section, we briefly explain the working principle of BatchNorm layer. Broadly speaking, the BatchNorm layer has been introduced to circumvent the issue of internal covariate shift (ICS). Consider a mini-batch $\mathcal{B}$ of size $M$, containing samples $\boldsymbol{x}_m$ for $m = 1, 2, ..., M$. BatchNorm normalizes the mini-batch during training, by calculating the mean $\mu_\beta$ and variance $\sigma_\beta^2$ as follows:

$$\mu_\mathcal{B} = \frac{1}{M} \sum_{i=1}^{M} \boldsymbol{x}_i \quad ; \quad \sigma_\mathcal{B} = \sqrt{\frac{1}{M} \sum_{i=1}^{M} (x_i - \mu_\mathcal{B})^2 + \epsilon}. \tag{6}$$

Activation normalization is then performed as,

$$\hat{\boldsymbol{x}}_i = \frac{\boldsymbol{x}_i - \mu_\mathcal{B}}{\sigma_\mathcal{B}}. \tag{7}$$

To further compensate for the possible loss of representational ability of network, BatchNorm also learns per-channel linear transformation as:

$$\boldsymbol{y}_i = \gamma \hat{\boldsymbol{x}}_i + \mathcal{B}, \tag{8}$$

for trainable parameters $\gamma$ and $\beta$ that represent scale and shift, respectively. These parameters are learnt using the same procedure, such as stochastic gradient descent, as other weights in the network. During training, the model usually maintains the moving averages of mini-batch means and variances(a.k.a. tracking), and during inference, uses these tracked statistics in place of the mini-batch statistics. Formally, tracking (moving average) of mean and variance, for scalar $\tau$ are given as,

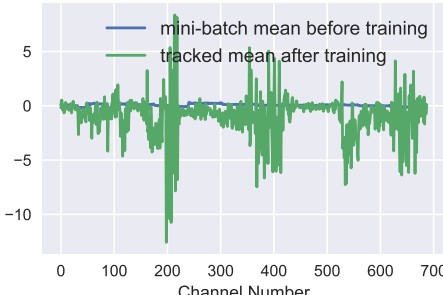 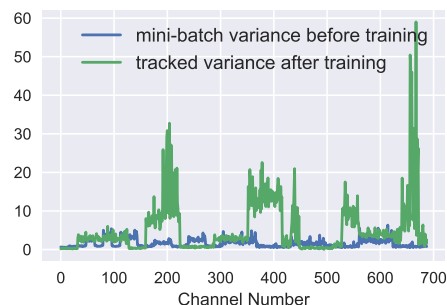

Figure 1: Comparison of mini-batch mean and variance at the start of the training with tracked mean and variance which are used at inference for all the channels. Note the difference between both of these values.

$$\hat{\mu} = (1 - \tau)\hat{\mu} + \tau\mu_\beta, \quad \hat{\sigma}^2 = (1 - \tau)\hat{\sigma}^2 + \tau\sigma_{\mathcal{B}}^2 \tag{9}$$

For inference, we can write,

$$\boldsymbol{y}_{\text{test}} = \frac{\boldsymbol{x}_{\text{test}} - \hat{\mu}}{\hat{\sigma}}.\gamma + \beta. \tag{10}$$

## 3.2 BATCHNORM AND ADVERSARIAL VULNERABILITY

Recently, Galloway et al. (2019) empirically showed that the accelerated training properties and occasionally higher clean test accuracy of employing BatchNorm in network come at the cost of low robustness to adversarial perturbations. While removing the BatchNorm layer may be helpful for robustness, it also means the loss of desirable properties of BatchNorm like very high learning rate, faster convergence, boost in test accuracy, etc. Therefore, it is pertinent to devise a normalization method that is not only robust to the adversarial perturbations but also inherits the desirable properties of the BatchNorm layer.

## 3.3 DEVIL IS IN THE TRACKING

In this section, we aim to investigate the reasons behind the adversarial vulnerability of the BatchNorm layer on the following two grounds;

- We note that during training, mini-batch statistics are used to normalize activations as shown in Equations 6 and 7. Moving average of these statistics are also calculated during the training that is called *tracking* (shown in Equation 9). The *tracked* mean and variance are used in the inference step (Equation 10). In this way, different values for mean and variance are used during training and inference. We show this in Figure 1 where it is clear that batch statistics at the start of the training are very different from tracked values that are used at the inference.

- Our second observation is based on the recent work of (Ding et al., 2019; Jacobsen et al., 2019). These works shed light on the link of the distributional shift in input data and robustness. Specifically, Ding et al. (2019) showed that adversarial robustness is highly sensitive to change in the input data distribution and prove that even *a semantically-lossless shift on the data distribution could result in drastically different robustness for adversarially trained models.*

Based on the above two observations, we hypothesize that: *Different values of first-order statistics (mean and variance) are used in the normalization layer for training and inference. This means different internal*

| | Norm | Clean | Noise | Gradient Sign | BIM-$\ell_\infty$ | PGD-$\ell_\infty$ |
|---|---|---|---|---|---|---|
| Natural Training | BatchNorm | **92.05 $\pm$ 0.34** | 59.16$\pm$6.74 | 33.75 $\pm$ 0.99 | 25.43$\pm$ 2.37 | 29.73 $\pm$0.86 |
| | No Norm | 82.93 $\pm$ 0.64 | 66.96$\pm$4.03 | 41.81 $\pm$ 2.20 | 41.47$\pm$2.16 | 37.01$\pm$2.04 |
| | BatchNorm w/o Tracking | 89.23$\pm$ 1.43 | **77.65 $\pm$ 2.45** | **53.32$\pm$ 0.71** | **48.38 $\pm$ 1.40** | **48.54$\pm$ 0.91** |
| Adversarial Training | BatchNorm | 79.53 $\pm$ 9.16 | 67.38$\pm$10.41 | 49.91 $\pm$ 6.30 | 47.90 $\pm$ 5.41 | 46.75 $\pm$ 6.05 |
| | No Norm | 83.60 $\pm$ 1.08 | 73.75 $\pm$3.95 | 63.46 $\pm$ 1.61 | 58.29 $\pm$ 2.53 | 60.01 $\pm$ 1.54 |
| | BatchNorm w/o Tracking | **89.28 $\pm$ 1.58** | **84.36 $\pm$ 1.06** | **72.42 $\pm$ 1.04** | **68.60$\pm$0.91** | **69.30 $\pm$1.77** |

Table 2: Comparison of adversarial robustness of BatchNorm with BatchNorm without tracking and no normalization for CIFAR10 and Resnet20. The results are shown with 95% confidence interval calculated with 5 random restarts. Highlighted values show the best accuracy for that particular adversarial noise. While BatchNorm without Tracking has significantly higher robust accuracy compare to both BatchNorm and no norm, its clean accuracy is lower than BatchNorm. But BatchNorm w/o Tracking also retains its clean accuracy when adversarially trained. This figure confirms our tracking-robustness hypothesis.

*representations being used at training and inference time which causes drift in input distributions of these layers. Therefore, the tracking part is the main culprit behind the adversarial vulnerability of the BatchNorm layer.*

To prove our hypothesis, we have done extensive experiments. For each experiment, we train a neural network model with three different normalization layers: BatchNorm, BatchNorm without tracking, and no normalization. To prove the generality of our argument, we have used various architectures, depths, and datasets as written in section 2.1. We train these networks on clean images as well as with based adversarial training procedure. For adversarial training, we use PGD attack for perturbation. We choose PGD due to its ability to generalize well for other adversarial attacks.

Table 1 shows our results on Resnet20. For detailed experimental results on various architectures, depths and dataset with different attacks see Table 4, 5 in appendix. The results clearly show that while the elimination of BatchNorm and training at a very small learning rate can help increase robustness, it also reduces clean data accuracy (with BatchNorm). *More importantly, this proves our hypothesis that by removing tracking, we can increase the robustness of a neural network significantly.* By using BatchNorm without tracking, we also keep many benefits of BatchNorm. Unfortunately, by eliminating the tracking part of BatchNorm, clean accuracy of a network also reduces as compared to clean accuracy with BatchNorm. We tackle this issue in the next section.

## 4 ROBUST NORMALIZATION

Although alleviation of ICS was claimed reason for the success of BatchNorm, recently, Bjorck et al. (2018) have shown that BatchNorm works because it avoids activation explosion by repeatedly correcting all activations. For this reason, it is possible to train networks with large learning rates, as activations cannot grow uncontrollably and convergence becomes easier. On a different side, recent work on robustness has shown a connection between the removal of outliers in activations and robustness (Xie et al., 2019; Etmann et al., 2019). Based on these observations, we use min-max rescaling that is often employed in preprocessing. This is also useful in the elimination of outliers since it rescales the data to a specific scale, Minmax normalization is defined as;

$$y_i = \frac{x_i - \min(x_i)}{\max(x_i) - \min(x_i)}. \tag{11}$$

|  | Natural Training | | Adversarial Training | |
|---|---|---|---|---|
|  | CIFAR10 | CIFAR100 | CIFAR10 | CIFAR100 |
| BatchNorm with tracking | $92.11 \pm 0.40$ | $68.20 \pm 0.8$ | $79.53 \pm 9.16$ | $35.39 \pm 3.91$ |
| BatchNorm w/o tracking | $89.07 \pm 1.73$ | $62.28 \pm 0.79$ | $89.28 \pm 1.58$ | $61.73 \pm 4.17$ |
| RobustNorm with tracking | $91.97 \pm 0.30$ | $68.13 \pm 0.28$ | $85.15 \pm 5.37$ | $42.19 \pm 6.10$ |
| RobustNorm w/o tracking | $91.49 \pm 0.28$ | $67.41 \pm 0.66$ | $90.76 \pm 0.63$ | $64.95 \pm 0.72$ |

Table 3: Comparison of clean accuracy of BatchNorm with RobustNorm for both adversarial and natural training scenarios. RobustNorm's accuracy is better than BatchNorm when tracking is not used while its accuracy is same when tracking is used.

However, we experimentally found this layer to be less effective in terms of convergence. Considering the importance of mean (Salimans & Kingma, 2016), we modify this to;

$$y_i = \frac{x_i - \mu}{\max(x_i) - \min(x_i)}. \tag{12}$$

We empirically observe the effectiveness of Equation 12 over Equation 11 but the overall performance was still inadequate. During debugging, we found that Equation 12 suppress activations much stronger than BatchNorm. This can also be seen from Popoviciu's inequality (Popoviciu, 1935),

$$4\sigma^2 \leq (\max(x_i) - \min(x_i))^2. \tag{13}$$

Following Popoviciu's inequality, we introduce the hyperparameter $0 < p < 1$, that reduces the denominator in Equation 12,

$$y_i = \frac{x_i - \mu}{(\max(x_i) - \min(x_i))^p} \tag{14}$$

We experimentally found that $p = 0.2$ value generalizes well for many networks as well as datasets. We call this normalization Robust Normalization (RobustNorm or RN) due to its robustness properties. We do not use tracking for the RobustNorm. But for comparison purposes, we keep running average of both mean and denominator and use this running average during inference and call this normalization RobustNorm with tracking.

Table 3 shows the accuracy of RobustNorm on Resnet20 for clean as well as adversarial training with both CIFAR10 and CIFAR100 datasets with 95% confidence interval calculated over 5 random restarts. These results show a better clean accuracy of RobustNorm in both natural and adversarial training scenarios. Apart from this, RobustNorm with tracking also shows better performance compared to BatchNorm with tracking. For adversarial robustness, we have shown Figure 2 with different attacks on CIFAR100 dataset. From the Table 3 and Figure 2, it is clear that RobustNorm keeps its clean accuracy while being more robust. For more results on Resnet38, Resnet50, VGG11, and VGG16 with CIFRAR10 and CIFAR100 datasets and both natural and adversarial training and many attack methods, please have a look at Table 4 and 5.

Figure3 shows the evolution of validation loss and accuracy for PGD based adversarial training with a confidence interval of 95% on Resnet20 architecture and CIFAR100 dataset. From Figure 8 in the appendix, it can also be seen that the evolution of training loss and accuracy is normal. But validation loss and accuracy for normalizations with tracking is much different for different random restarts. This can probably be explained based on flat and sharp minima attained by different normalizations as can be seen in loss landscape in Figure 9 in the appendix. For further discussion on the loss landscape, please see section B in appendix.

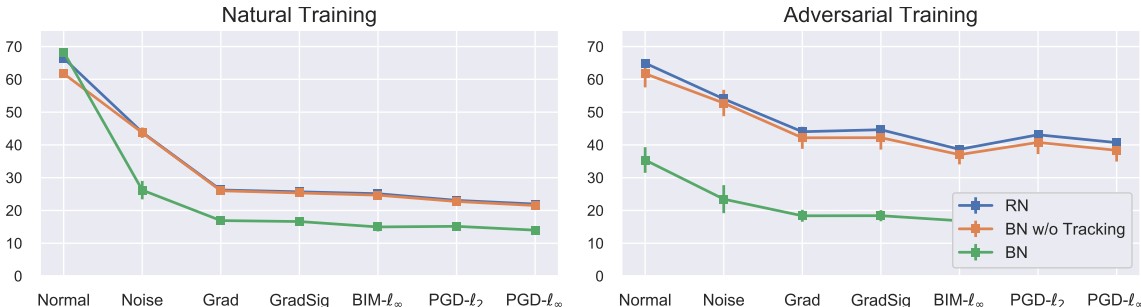

Figure 2: Comparison of adversarial robustness of different normalizations for different Whitebox attacks with CIFAR100. Results are shown with 95% confidence interval computed over 5 random restarts. Batch-Norm w/o Tracking and RobustNorm have significantly higher adversarial robustness compared to other norms. The results are even more clear when they are adversarially trained where RobustNorm's robustness is even more than BatchNorm w/o Tracking.

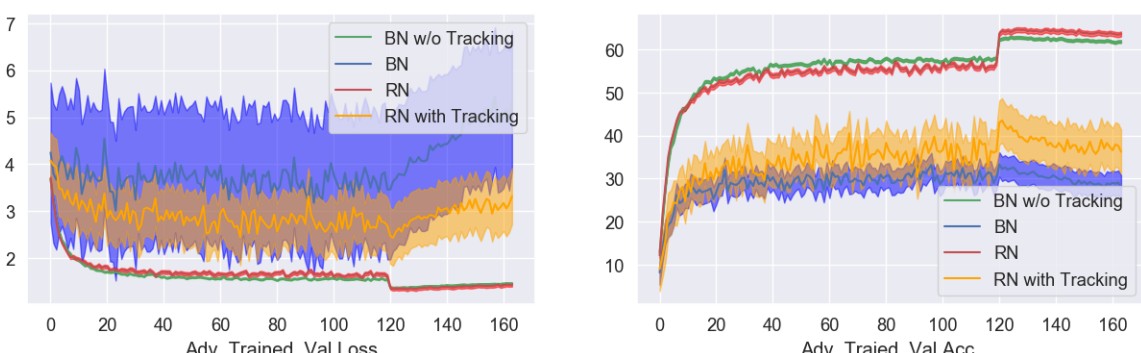

Figure 3: Comparison of evolution of validation loss and accuracy for CIFAR100 on Resnet20 with confidence intervals calculated with 5 random restarts. BatchNorm without tracking and RobustNorm have higher accuracy and lower loss while RobustNorm being better than BatchNorm w/o tracking. For further details, please have a look at appendix B

### 4.1 RESISTANCE FOR DIFFERENT VALUES OF ADVERSARIAL PERTURBNESS

To further understand the performance of RobustNorm under adversarial conditions, we run an experiment where $\epsilon$ values are increased for the test set. We train networks with BatchNorm, BatchNorm w/o Tracking and RobustNorm with Natural as well as PGD-$\ell_\infty$ based adversarial training and tested them on different values of $\epsilon$. The results are shown in Figure 4. As $\epsilon$ increases, the robustness of neural network decreases but the robustness of neural network with RobustNorm is much higher than BatchNorm while also higher than BatchNorm w/o tracking. To see the effect of an increase in adversarial noise on CIFAR100 dataset, see Figure 6 in the appendix.

## 5 IS TRACKING A NECESSARY EVIL?

In the previous sections, we have empirically shown the wickedness of tracking in BatchNorm. But there is more to the story. One benefit of tracking that makes it a necessary evil in BatchNorm is its ability to have

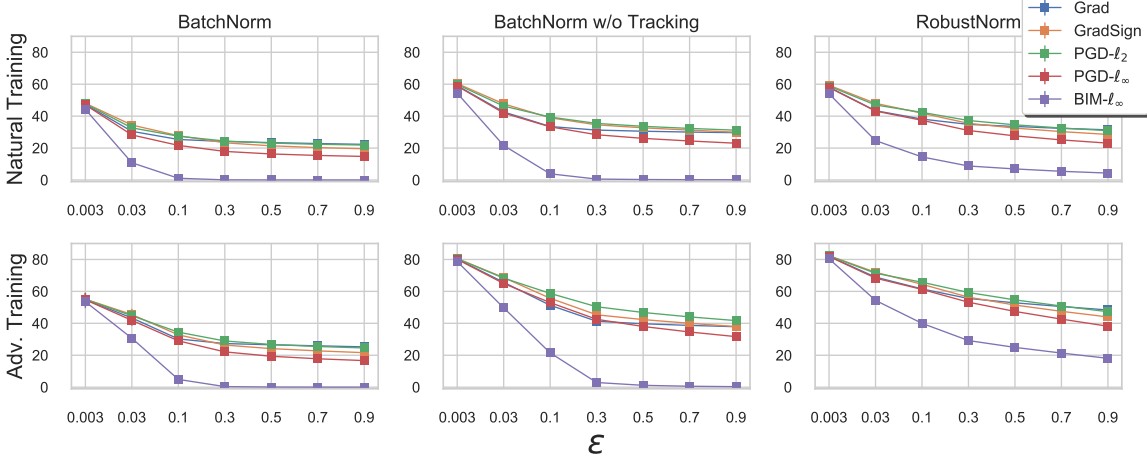

Figure 4: Comparison of robust accuracy as $\epsilon$(adversarial noise power) increases. Values are calculated for CIFAR10 with 3 random restarts and confidence interval of 95%. Robustness of RN is higher than both BatchNorm and BatchNorm without tracking. While BatchNomr with and without tracking collapses with higher epsilon BIM, RobustNorm's accuracy is much higher. For same curves on CIFAR10, see appendix for more details.

consistent results on small inference time batch sizes. For all the results mentioned in this paper, we have used 100 as inference time batch size. By removing tracking, we lose this ability as shown in Figure 5. Does this make tracking a necessary evil? Based on the observations in section 3, one straight forward solution to this problem could be the use of tracked values during training as well since it will change the issue of different distributions. This, however, causes the model to blow up as argued by Ioffe & Szegedy (2015); Ioffe (2017). To resolve this issue, one possible solution is Batch Renormalization like formulation (Ioffe, 2017) but it requires careful tuning of many hyperparameters which makes its use very difficult. Similarly, other possible solutions such as Arpit et al. (2016); Salimans & Kingma (2016); Wu & He (2018) have their own challenges as well. From these observations, we argue that this problem requires more investigation.

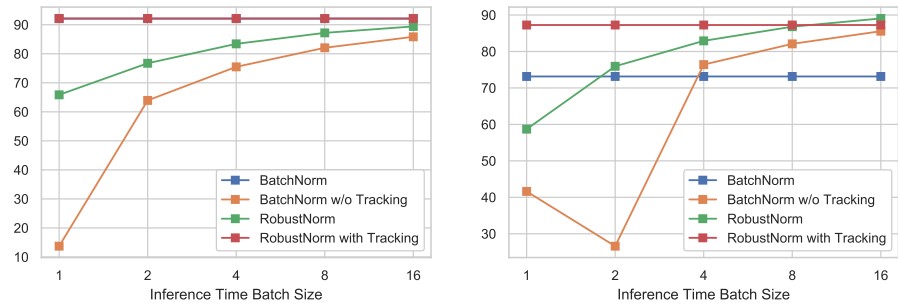

Figure 5: Effect of using very small inference time batch size on different norms. Part(a) shows results for natural training while part(b) shows results for adversarial trained networks.

It is also important to note that RobustNorm's results are restored by increasing batch size by a small number. Similarly, tracking is also less harmful for RobustNorm and RobustNorm with tracking is still more robust while having all the benefits of BatchNorm.

## 6 CONCLUSION

Addition of maliciously crafted noise in normal inputs, also called adversarial examples has proven to be deceptive for neural networks. While there are many reasons for this phenomena, recent work has shown BatchNorm to be a cause of this vulnerability as well. In this paper, we have investigated the reasons behind this issue and found that tracking part of BatchNorm causes this adversarial vulnerability. Then, we showed that by eliminating it, we can increase the robustness of a neural network. Afterward, based on the intuitions from the work done for the understanding of BatchNorm, we proposed RobustNorm which has much higher robustness than BatchNorm for both natural as well as adversarial training scenarios. In the end, we have shown how tracking can be a necessary evil and argued that it requires further careful investigation.

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

# A  ROBUSTNORM

## A.1  ROBUSTNORM FOR OTHER ARCHITECTURES

In this section, we provide more detailed results for our experiments for both CIFAR10 and CIFAR100 datasets. Table 4 and 5 shows detailed results.

| | Norm | Clean | Noise | GradSign | BIM-linf | PGD-$l$ inf |
|---|---|---|---|---|---|---|
| Natural Training Resnet38 | BatchNorm | 93.11 | 62.67 | 41.53 | 36.88 | 35.12 |
| | BatchNorm w/o tracking | 90.62 | 79.42 | 56.80 | 51.31 | 50.68 |
| | RobustNorm | 92.24 | 80.35 | 58.37 | 55.01 | 53.69 |
| | RobustNorm with Tracking | 92.71 | 62.31 | 43.46 | 38.33 | 37.45 |
| Adversarial Training Resnet38 | BatchNorm | 73.21 | 59.21 | 43.54 | 42.06 | 39.77 |
| | BatchNorm w/o Tracking | 89.74 | 84.82 | 75.21 | 69.45 | 71.91 |
| | RoubstNorm | 91.77 | 85.57 | 77.29 | 72.23 | 74.23 |
| | RoubstNorm with Tracking | 87.08 | 72.86 | 58.42 | 54.56 | 54.09 |
| Natural Training Resnet50 | BatchNorm | 93.61 | 60.44 | 42.06 | 36.15 | 35.71 |
| | BatchNorm w/o Tracking | 90.38 | 77.88 | 57.36 | 51.18 | 51.44 |
| | RoubstNorm | 91.67 | 79.33 | 57.01 | 53.12 | 51.75 |
| | RoubstNorm with Tracking | 93.23 | 66.14 | 44.88 | 40.33 | 38.50 |
| Adversarial Training Resnet50 | BatchNorm | 65.99 | 52.33 | 30.92 | 35.42 | 28.40 |
| | BatchNorm w/o Tracking | 90.22 | 86.29 | 78.17 | 73.01 | 75.21 |
| | RoubstNorm | 90.91 | 86.14 | 78.03 | 73.29 | 75.21 |
| | RoubstNorm with Tracking | 63.12 | 50.68 | 43.37 | 39.45 | 40.36 |
| Natural Training VGG11 | BatchNorm | 91.66 | 81.80 | 51.71 | 54.07 | 46.66 |
| | BatchNorm w/o Tracking | 90.28 | 86.70 | 64.29 | 62.86 | 59.97 |
| | RoubstNorm | 90.51 | 86.69 | 29.53 | 48.69 | 25.71 |
| | RoubstNorm with Tracking | 91.74 | 79.53 | 52.35 | 52.65 | 47.04 |
| Adversarial Training VGG11 | BatchNorm | 79.18 | 70.48 | 52.63 | 52.38 | 48.99 |
| | BatchNorm w/o Tracking | 89.69 | 86.78 | 76.98 | 73.10 | 74.51 |
| | RoubstNorm | 90.81 | 87.81 | 77.53 | 74.2 | 74.84 |
| | RoubstNorm with Tracking | 0 | 0 | 0 | 0 | 0 |
| Natural Training VGG16 | BatchNorm | 93.56 | 66.68 | 49.83 | 44.52 | 44.38 |
| | BatchNorm w/o Tracking | 92.44 | 84.56 | 59.64 | 54.35 | 52.81 |
| | RoubstNorm | 92.62 | 85.52 | 45.01 | 52.01 | 39.02 |
| | RoubstNorm with Tracking | 93.54 | 73.86 | 50.98 | 48.52 | 45.43 |
| Adversarial Training VGG16 | BatchNorm | 82.27 | 71.77 | 56.78 | 53.58 | 52.61 |
| | BatchNorm w/o Tracking | 92.06 | 87.52 | 76.49 | 71.48 | 72.98 |
| | RoubstNorm | 92.53 | 88.32 | 80.65 | 77.65 | 78.70 |
| | RoubstNorm with Tracking | 91.20 | 75.39 | 68.66 | 65,28 | 65.34 |

Table 4: Comparison of clean and adversarial accuracy of different network architectures. We present two depths of Resnet, 38 and 50 and two for VGG, 11 and 16. We trained all these networks with both natural as well as adversarial training and use noise as well as different attacks methods to find their robustness. All of these results are for CIFAR10 dataset.

|  | Norm | Clean | Noise | GradSign | BIM-linf | PGD-$l$ inf |
|---|---|---|---|---|---|---|
| Natural Training Resnet38 | BatchNorm | 70.22 | 31.43 | 18.77 | 18.01 | 15.88 |
|  | BatchNorm w/o Tracking | 63.9 | 46.65 | 27.03 | 26.38 | 23.06 |
|  | RoubstNorm | 68.16 | 47.88 | 28.12 | 27.98 | 24.27 |
|  | RoubstNorm with Tracking | 70.46 | 26.79 | 17.91 | 15.80 | 15.20 |
| Adversarial Training Resnet38 | BatchNorm | 42.54 | 30.78 | 24.63 | 21.88 | 22.33 |
|  | BatchNorm w/o Tracking | 65.71 | 57.70 | 46.46 | 41.47 | 42.68 |
|  | RoubstNorm | 67.91 | 57.39 | 48.03 | 42.18 | 44.14 |
|  | RoubstNorm with Tracking | 43.44 | 26.29 | 28.49 | 21.01 | 25.99 |
| Natural Training Resnet50 | BatchNorm | 73.67 | 29.18 | 18.19 | 17.15 | 15.55 |
|  | BatchNorm w/o Tracking | 66.90 | 48.11 | 27.25 | 27.06 | 23.16 |
|  | RoubstNorm | 67.88 | 46.04 | 28.60 | 27.98 | 25.17 |
|  | RoubstNorm with Tracking | 72.79 | 31.67 | 17.72 | 17.90 | 14.92 |
| Adversarial Training Resnet50 | BatchNorm | 25.00 | 16.01 | 14.21 | 12.25 | 13.11 |
|  | BatchNorm w/o Tracking | 66.55 | 59.06 | 49.31 | 44.10 | 45.79 |
|  | RoubstNorm | 67.24 | 58.14 | 50.87 | 45.46 | 47.77 |
|  | RoubstNorm with Tracking | 24.32 | 13.38 | 17.75 | 12.77 | 16.33 |
| Natural Training VGG11 | BatchNorm | 69.93 | 50.97 | 26.88 | 30.56 | 23.53 |
|  | BatchNorm w/o Tracking | 67.62 | 60.49 | 36.30 | 37.99 | 32.66 |
|  | RoubstNorm | 68.77 | 61.35 | 16.18 | 31.21 | 13.16 |
|  | RoubstNorm with Tracking | 69.96 | 48.29 | 25.89 | 28.90 | 22.63 |
| Adversarial Training VGG11 | BatchNorm | 47.69 | 28.78 | 28.56 | 23.50 | 26.56 |
|  | BatchNorm w/o Tracking | 66.12 | 61.33 | 49.82 | 46.85 | 47.06 |
|  | RoubstNorm | 68.02 | 61.40 | 50.68 | 47.09 | 47.57 |
|  | RoubstNorm with Tracking | 64.43 | 43.31 | 44.67 | 41.67 | 41.77 |
| Natural Training VGG16 | BatchNorm | 73.28 | 36.12 | 22.49 | 21.74 | 19.49 |
|  | BatchNorm w/o Tracking | 70.34 | 56.15 | 33.97 | 33.36 | 29.46 |
|  | RoubstNorm | 72.28 | 56.33 | 16.65 | 28.87 | 13.53 |
|  | RoubstNorm with Tracking | 73.40 | 38.31 | 25.43 | 24.93 | 23.78 |
| Adversarial Training VGG16 | BatchNorm | 53.11 | 26.39 | 31.23 | 31.21 | 28.89 |
|  | BatchNorm w/o Tracking | 69.33 | 61.49 | 51.16 | 46.83 | 48.19 |
|  | RoubstNorm | 71.01 | 61.89 | 53.07 | 48.89 | 50.39 |
|  | RoubstNorm with Tracking | 68.34 | 39.22 | 43.89 | 34.10 | 41.07 |

Table 5: Comparison of clean and adversarial accuracy of different network architectures for CIFAR100. We present two depths of Resnet, 38 and 50 and two for VGG, 11 and 16. We trained all these networks with both natural as well as adversarial training and use noise as well as different attacks methods to find their robustness.

## A.2 RESISTANCE TO INCREASE IN ADVERSARIAL NOISE

In this section, we put results of increasing adversarial noise on CIFAR100 dataset.

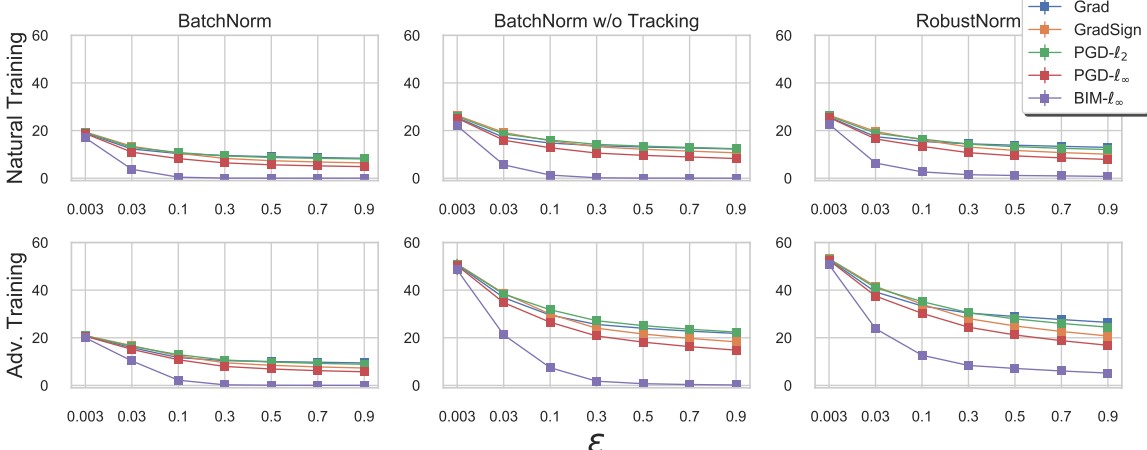

Figure 6: Figure shows effect of increase in adversarial noise $\epsilon$ on three normalizatons for CIFAR100 dataset.

### A.3 ANOTHER ASPECT OF ROBUSTNORM

As we have discussed, ICS hypothesis has been negated by a few recent studies. One of these studies (Santurkar et al., 2018) suggested that based on the results, " it might be valuable to perform a principled exploration of the design space of normalization schemes as it can lead to better performance." In this way, we can see RobustNorm with tracking as a new normalization scheme which is based on alternative explanations yet having performance equal to BatchNorm which, in a way, weakens ICS hypothesis. See Figure 7 for a comparison of accuracies over different models for CIFAR10 and CIFAR100 datasets.

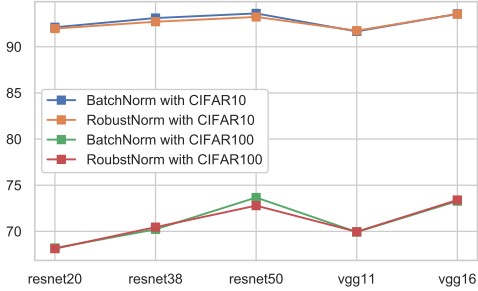

Figure 7: Comparison of BatchNorm with RobustNorm in terms of accuracy when tracking is used. Both of these norms have very similar clean accuracy despite RobustNorm being different in terms of ICS hypothesis.

## B   LOSS LANDSCAPE

In this section, we discuss possible reasons for the high variation of validation loss for a network with normalization with tracking. As shown in Figure 8, training loss of all the norms converges with similar

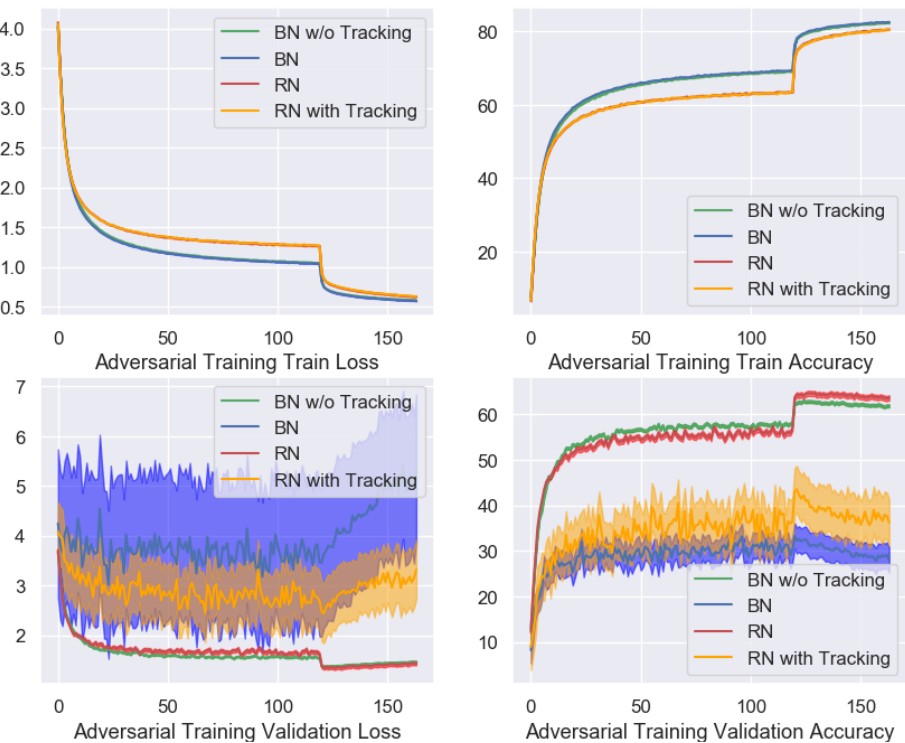

Figure 8: Training and validation loss and accuracy evolution for adversarial training. Interestingly, Batch-Norm's training loss decreases normally, its validation loss has lot more uncertainty and either remaining same or start increasing. This way, BatchNorm is overfitting. A similar trend is also shown by RobustNorm when tracking is used thought it vanishes when remove tracking and decrease in loss becomes normal and less uncertain.

fashion for all the random restarts but validation loss has a lot of variation over these restarts. In other words, the value of training loss is similar among many restarts while the same loss values change drastically for validation. To further understand it, we draw loss landscape of these networks using formulation given by Li et al. (2018b) in Figure 9. From these plots, we observe an interesting behaviour: networks having normalization without tracking(i.e. better robustness) tend to have sharp minima as can be seen in figures 9c, 9d, 9e, 9f while their counterparts have more flat loss landscape i.e. figures 9a, 9b, 9g, 9h. There is a long history of debate on generalization ability of sharp vs flat minima Hochreiter & Schmidhuber (1997); Keskar et al. (2016); Dinh et al. (2017). We think more work in this direction can lead to a better understanding of how BatchNorm causes this vulnerability.

## C    MORE EXPERIMENTS ON ROBUSTNORM

In this section, we discuss some less interesting experiments done to understand the role of power in Ro-bustNorm.

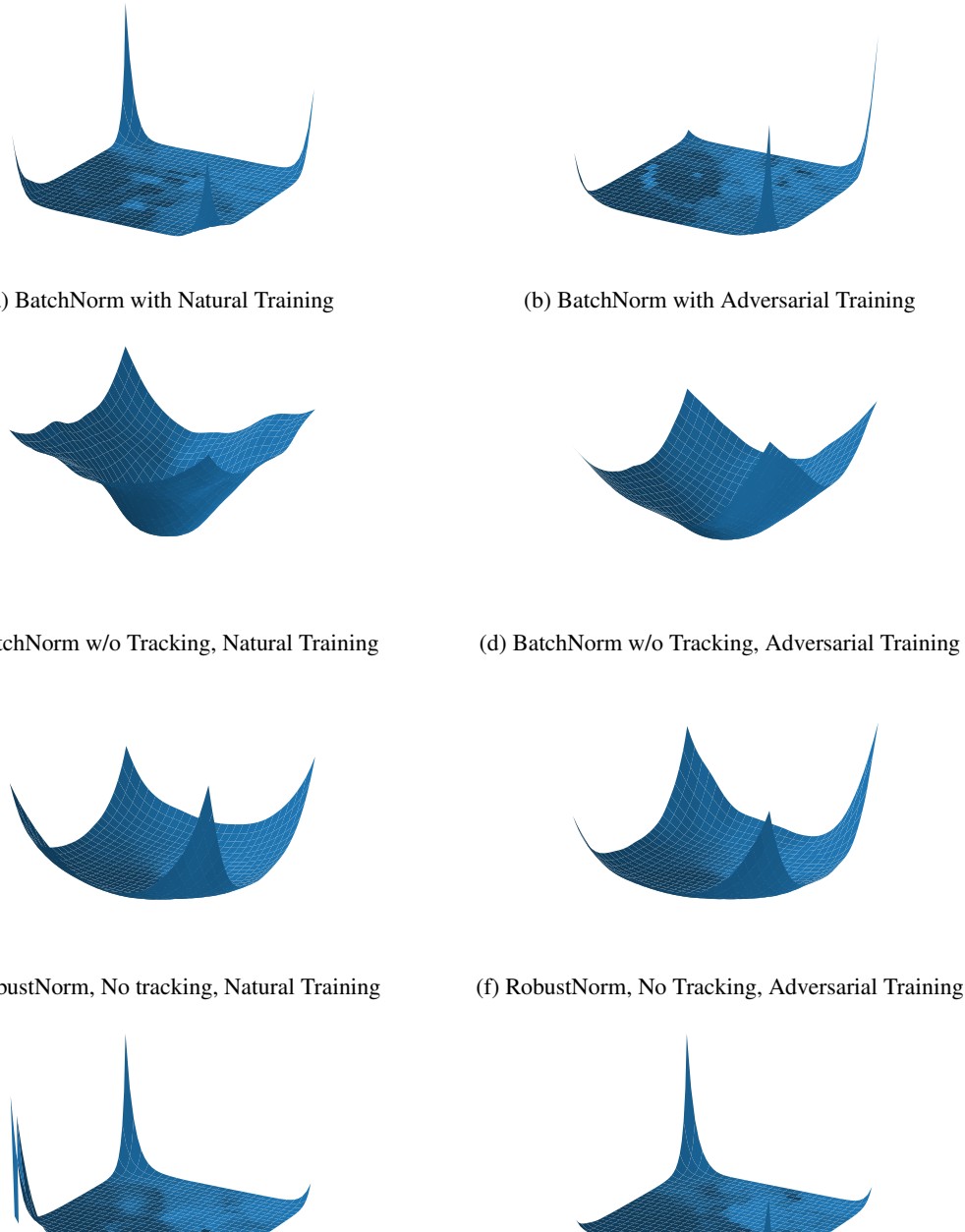

(a) BatchNorm with Natural Training

(b) BatchNorm with Adversarial Training

(c) BatchNorm w/o Tracking, Natural Training

(d) BatchNorm w/o Tracking, Adversarial Training

(e) RobustNorm, No tracking, Natural Training

(f) RobustNorm, No Tracking, Adversarial Training

(g) RobustNorm, with Tracking, Natural Training

(h) RobustNorm, with Tracking, Adversarial Training

Figure 9: Plots of loss landscape (Li et al., 2018a) of Resnet20 trained with different norms. Right column shows Resnet20 trained on natural images while left column shows adversarially trained. Note that network without tracking (9c, 9d, 9e, 9f) tends to have sharp minima.

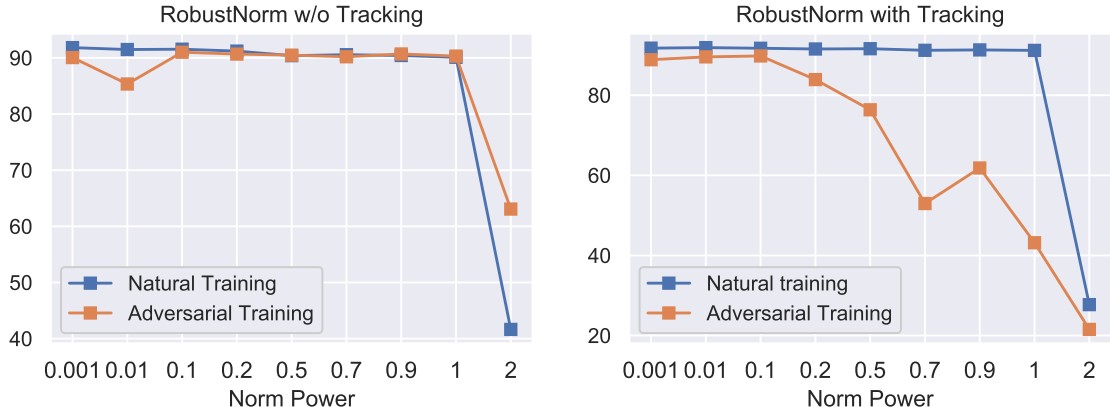

Figure 10: Comparison of effect of hyperparameter $p$ on clean accuracy for both adversarially trained and naturally trained Resnet20 with RobustNorm with and without tracking.

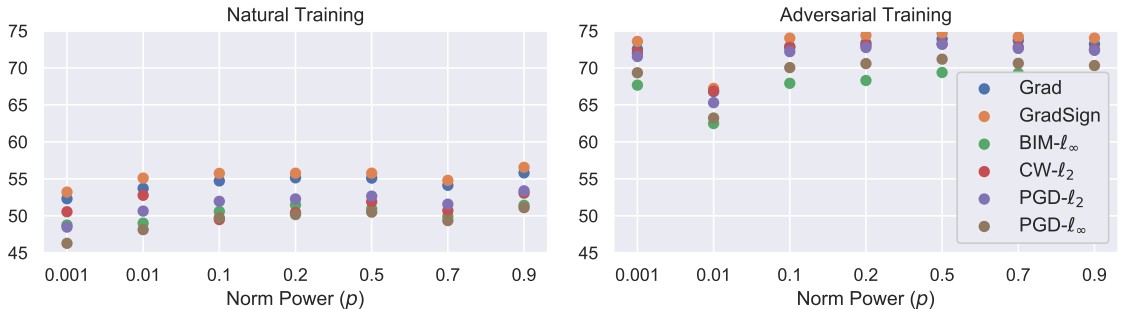

Figure 11: Comparison of effect of hyperparameter $p$ on clean accuracy.

## C.1 EFFECT OF POWER ON ROBUSTNORM

In this section, we show the effect of changing hyperparameter $p$ for clean as well as robust accuracy. From figure 4 and 11, it can be seen that both robustness to many attacks as well as accuracy changing with the power. So it can be concluded that by tuning hyperparameters, we can get better results.

