# OpenReview forum: "Towards an Adversarially Robust Normalization Approach"
_ICLR.cc/2020/Conference — Reject_

### Official Review · AnonReviewer2 · 2019-10-22
**Official Blind Review #2**

**Rating:** 3

**Review:**

This paper addresses a limitation of BatchNorm: vulnerability to adversarial perturbations. The authors propose a possible explanation of this issue and correspondingly an alternative called RobustNorm to tackle this problem. Specifically, the authors observe that the statistics of BatchNorm for training and inference are different, resulting in different data distributions for training and inference. To solve this problem, the authors propose to use min-max rescaling instead of normalization. In addition, the running average is calculated with mean and the running mean of the denominator during inference. Experimental results show significant improvement of robustness and also comparable accuracy for clean data.

The paper is well-written and the contributions are stated clearly. The explanation of vulnerability is reasonable. The proposed solution is simple but effective.

However, I have several concerns:
*The authors verify that the running average is the main culprit of vulnerability to adversarial attack, but provide no further investigation of why this happens. A possible solution is the drift in input distributions, but the manuscript does not state clearly how is the distribution changed. Further experiments would have made this claim more convincing.
*The proposed method involves a hyper-parameter \rho, but it may result in problematic issues. The variance of input is of the same order of magnitude as (max(x)-min(x))^2. If \rho is set to other value, the magnitude of gradient will change drastically during back-propagation. Although \rho can be set to 0.2, it still seems ad-hoc. Experiments on more datasets and the sensitivity of the proposed method to \rho would have validated the claims of the authors.


**Experience Assessment:**

I have read many papers in this area.

**Review Assessment: Checking Correctness Of Derivations And Theory:**

I carefully checked the derivations and theory.

**Review Assessment: Checking Correctness Of Experiments:**

I assessed the sensibility of the experiments.

**Review Assessment: Thoroughness In Paper Reading:**

I read the paper thoroughly.

---

### Official Review · AnonReviewer1 · 2019-10-23
**Official Blind Review #1**

**Rating:** 3

**Review:**

Review: This paper investigates the reason behind the vulnerability of BatchNorm and proposes a Robust Normalization. They experimentally show that it is the moving averages of mini-batch means and variances (tracking) used in Normalization that cause the adversarial vulnerability. Based on this observation, they propose a new normalization method not only achieves significantly better results under a variety of attack methods but ensures a comparable test accuracy to that of BatchNorm on unperturbed datasets. The paper is clearly written, easy to read.

Strengths:

Explore the cause of adversarial vulnerability of the BatchNorm and assume that the tracking mechanism used in original BatchNorm leads to the vulnerability from experiment results.
Propose a new and simple normalization method and perform extensive experiments to validate the efficacy of proposed method.

Weaknesses:
Though extensive experiments have been done by revealing what leads the vulnerability and the effectiveness of proposed method. The results seem unconvincing with respect to different datasets, since Cifar10 and Cifar100 are inherently connected. Would you mind performing some experiments on ImageNet? Since adversarial training on ImageNet is time-consuming, can you show us the result of Natural Training of different models with different norms on ImageNet and compare their robustness under different attack?


**Experience Assessment:**

I have published one or two papers in this area.

**Review Assessment: Checking Correctness Of Derivations And Theory:**

I assessed the sensibility of the derivations and theory.

**Review Assessment: Checking Correctness Of Experiments:**

I assessed the sensibility of the experiments.

**Review Assessment: Thoroughness In Paper Reading:**

I read the paper at least twice and used my best judgement in assessing the paper.

---

### Official Review · AnonReviewer3 · 2019-10-23
**Official Blind Review #3**

**Rating:** 6

**Review:**

This paper proposes an interesting perspective that BatchNorm may introduce the adversarial vulnerability, and probes why BatchNorm performs like that (the tracking part in BatchNorm). In experiment, the robustness of the networks increases by 20% when removing the tracking part, but the test accuracy on the clean images drops a lot. Afterwards, the authors propose RobustNorm, which performs better than BatchNorm for both natural and adversarial scenarios.

Detailed Comments:
+ The paper is well written. The paper structure is clear and figures are well illustrated.
+ The paper understands and carefully investigates BatchNorm in a interesting and important direction. After the investigation, the improved version RobustNorm shows more potential.
+ The experimental results seem good. The RobustNorm performs better than BatchNorm for both natural and adversarial scenarios.
- More results on ImageNet would be better to verify the proposed RobustNorm method.

**Experience Assessment:**

I have read many papers in this area.

**Review Assessment: Checking Correctness Of Derivations And Theory:**

I assessed the sensibility of the derivations and theory.

**Review Assessment: Checking Correctness Of Experiments:**

I carefully checked the experiments.

**Review Assessment: Thoroughness In Paper Reading:**

I read the paper at least twice and used my best judgement in assessing the paper.

---

### Public Comment · ~Angus_Galloway1 · 2019-09-27
**A few questions**

The results of Table 2 are very nice, showing that the degradation in robustness caused by BatchNorm persists for PGD adversarial training when evaluated on the same. We did not include this in the original work, but have since found similar results by a slimmer margin with the WideResNet architecture from Madry et al. I would maintain however as an aside that robustness also be assessed on unseen attacks and corruptions.

One concern I have here is that the clean test accuracy of the ResNet20 without BatchNorm is 9-10 points lower than with BatchNorm. Are you using Fixup initialization as the unnormalized baseline (https://openreview.net/forum?id=H1gsz30cKX)? I find that unnormalized ResNets can outperform their batch-normalized equivalent in terms of clean test accuracy, even when using training hyperparameters that were originally tuned specifically for batch-normalized models. For example, here are some ResNet110 checkpoints, the Fixup variant gets ~93.0 vs ~92.5 for BatchNorm: https://github.com/AngusG/bn-advex-zhang-fixup.

Misc questions:
- How is the data preprocessed for various experiments in this work?
- What was the value of the normalization constant for BatchNorm (We have since realized that this is an important hyperparameter of BatchNorm, but seldom reported. Pytorch defaults to 1e-5, tf to 1e-3).
- Apologies if I missed this, what value of \tau is used in the experiments with tracking?
- In Figure 4, accuracy plateaus rather than reaching zero as epsilon increases up to 0.9. This suggests vanishing gradients. What was the attack objective here? Sometimes switching from misclassification to a targeted objective can recover the efficacy of a white-box attack. I noticed exactly this for PGD max-norm trained ResNets with Fixup when initially evaluating PGD 2-norm robustness; the misclassification objective would plateau, but targeted attacks go to zero.

---

> ### Author Response · Authors · 2019-10-08
> **Re: A few questions**
>
> Thank you for your comment. Here we will answer your question one by one.
>
> - ``"however, as an aside, that robustness also be assessed on unseen attacks and corruptions". By unseen attacks and corruption, do you mean attacks other than PGD on which we have adversarially trained? If so, yes, we have included results for many attacks and Gaussian noise. For instance, table 2 shows the effect of Noise, GradientSign, BIM-$\ell_{\infty}$ and PGD-$\ell_{\infty}$. Similarly, figure 4 shows results for two additional attacks, Grad and PGD-$\ell_{2}$ along with different $\epsilon$ levels ranging from 0.003/1 to 0.9/1. Similarly, other results also show the effect of unseen attacks.
>
> If by unseen attacks, you mean BlackBox settings, we also have tested one such setting based on transferability property [6]. For this experiment, we have crafted iterative PGD-$\ell_{\infty}$ noise for Renet20 with the BatchNorm layer and used it for ResNet38 with different normalizations. Please note that for the creation of adversarial samples, we have used only the BatchNorm layer to comply with BlackBox settings. As suggested by [7], we have only used a non-targeted attack. We will add more results in the final version of the paper.
>
> -----------------------------------------------------------------------------------
>                           BN                      BN w/o Tracking            RN
> -----------------------------------------------------------------------------------
> Renset38         43.96                   47.12                               52.71
> ------------------------------------------------------------------------------------
>
> -   For a fair comparison, we used the same network with the same settings except for the normalization layer. This means, for the un-normalized network, we removed the BatchNorm layer. We did not use fixup initialization and used standard initialization used by most of the researchers. We agree that fixup initialization can increase accuracy in some cases and can eliminate the use of the BatchNorm layer. Although we have not experimented with fixup initialization, based on the experiments in the paper [5], it can be helpful for the reduction in adversarial vulnerability. But according to the Fixup paper, we also need to use multiplier and bias factors and some kind of regularization to match the accuracy of BatchNorm.
>
> Also, this, in a way, supports our hypothesis that tracking is a cause of adversarial vulnerability.  Because fixup initialization solves the problem (mysterious problem with many interpretations such as alleviation of internal Covariate  shift(ICS) [4], correction of activations to avoid their explosion [1], making loss landscape smoother [2] or regularization [3], etc.) in a way which do not make input distribution at train and test time different.
>
> - For a fair comparison, we have used uniform settings across our experiments. For data preprocessing, we have used random crops with padding of 4 and random horizontal flips. Similarly, we have used Pytorch default values for all variables which means eps = 1e-5 and tau = 0.1. Please note that we have not tuned these settings for our experiments to get better results.
>
> - We do agree with your observation. By switching from misclassification to targeted objective, adversarial accuracy improves. We have used the misclassification objective for all of our experiments. Thank you for highlighting it, We will add relevant experiments in the appendix. Yes, accuracy with BIM-$\ell_{\infty}$ attack decreases to zero for BatchNorm as well as BatchNorm w/o tracking as the value of $\epsilon$ is increased to a very high level. For other attacks, this is not the case. Also, note that RobustNorm is still resistive to the attacks with such high levels of adversarial noise.
>
>
> [1] Bjorck, Nils, et al. "Understanding batch normalization." Advances in Neural Information Processing Systems. 2018.
>
> [2] Santurkar, Shibani, et al. "How does batch normalization help optimization?." Advances in Neural Information Processing Systems. 2018.
>
> [3] Luo, Ping, et al. "Towards understanding regularization in batch normalization." (2018).
>
> [4] Ioffe, Sergey. "Batch renormalization: Towards reducing minibatch dependence in batch-normalized models." Advances in neural information processing systems. 2017.
>
> [5] Galloway, Angus, et al. "Batch Normalization is a Cause of Adversarial Vulnerability." arXiv preprint arXiv:1905.02161 (2019).
>
> [6] Papernot, Nicolas, Patrick McDaniel, and Ian Goodfellow. "Transferability in machine learning: from phenomena to black-box attacks using adversarial samples." arXiv preprint arXiv:1605.07277 (2016).
>
> [7] Liu, Yanpei, et al. "Delving into transferable adversarial examples and black-box attacks." arXiv preprint arXiv:1611.02770 (2016).

---

### Author Response · Authors · 2019-10-08
**Litmus Test of Paper's Hypothesis**

As suggested by Anthony Wittmer in the comments section of the paper [1], "BatchNorm is the cause of adversarial vulnerability",  a litmus test for our hypothesis, "tracking in BatchNorm is a cause of its adversarial vulnerability", is to test it for normalizations that do not require tracking. We have tested this for Group and LayerNorm. Both of these normalizations don't require tracking for test time statistics.  The results are shown in the following two figures. From figures, it is clear that normalization without tracking is less vulnerable to adversarial attacks. We believe this can also help solve the problem mentioned in section 5 of the paper.

https://ibb.co/B37LFFS
The figure shows the effect of different attacks on the accuracy of networks trained with different normalizations. All the normalizations that don't require have much better adversarial accuracy.

https://ibb.co/JQvYjjB
Effect of increasing $\epsilon$ for different adversarial attacks on networks with different norms. The first sub-figure shows results for BatchNorm with tracking and the rest of the figure shows results for normalizations without tracking.



[1] \url{https://openreview.net/forum?id=H1x-3xSKDr&noteId=SklNaNu2vr}

---

> ### Public Comment · ~Anthony_Wittmer1 · 2019-10-24
> **Thanks**
>
> Hi, thanks for the additional results.
>
> From the above results, it seems that both LayerNorm and GroupNorm show better robustness than BatchNorm (with or without tracking) . Have the authors tried to compare the RN with LayerNorm and GroupNorm? Does the result reveal that the normalizations on a single sample show better robustness than  the normalizations on the batch samples?
>
> From the result of Table 4, could the authors explain why natural training with RN  show worse performance on VGG11 and VGG16 for CIFAR10 and CIFAR100 ?

---

### Public Comment · ~Anthony_Wittmer1 · 2019-10-15
**Minor problem about the page format**

Hi,

I find that the page length is shorter than the standard length, i.e., there are a lot of blank lines under the page number of each page.

---

> ### Author Response · Authors · 2019-10-15
> **Thanks**
>
> Thank you for pointing this out.
> We have used standard ICLR latex template in overleaf. We will try to fix this in our final submission.
>
> We would love to hear more feedback from you.

---

### Public Comment · ~Anthony_Wittmer1 · 2019-10-23
**typos and the adversarial vulnerability of BN for TRADES**

In Section 2:

"2.0.1 ADVERSARIAL TRAINING:" => "2.0 ADVERSARIAL TRAINING"
"100" => "CIFAR100"
"Table 2" => "Table 1" . Table 1 is missing

Have the authors tried to evaluate the adversarial vulnerability of BN for TRADES[1], which combines the clean data and the adversarial data to train the nerual networks. Unlike PGD adversarial training, for the training process of TRADES, the normalization statistics for BN involve the clean data and the adversarial data, which may decrease the distance of  the normalization statistics during training and inference phase.

Is the performance of "BatchNom w/o tracking" or "RN"  related with the batch size in the inference phase, where the normalization statistics are related with the testing data. Sorry, I missing the results in Figure 5, where the result of BN is missing for Part a.



[1] Theoretically Principled Trade-off between Robustness and Accuracy. ICML 2019

---

### Public Comment · ~Jungeum_Kim1 · 2019-10-24
**Small questions**

Hello! I think your work is interesting and has practical values for applying adversarial training in real applications. Can I ask several questions to better understand your work?

First, for the model of BatchNorm w/o tracking, in the inference time, how do you normalize? Or, do you apply no batch normalization? Likewise, what about RobustNorm w/o tracking? I am assuming you applied some normalization that I did not realize because no normalization would again change the data distribution, which is one of main things that you want to preserve.

Second, in deriving RobustNorm with $\mu$ and $p=0.2$, you mention that "Equation 12 suppress activations much stronger than BatchNorm". What do you mean by suppress? May I get a little more explanation about your observation on Equation (12) w.r.t. this sentence?

Third, the results in Figure 4 for PGD-$\ell_\infty$ is strange. In my understanding, BIM in your (2) is equivalent to PGD-$\ell_\infty$ without initial random noise. However, in Figure 4, the performance of PGD-$\ell_\infty$ is significantly worse than BIM-$\ell_\infty$ about by 10% to 30%. Moreover, in your table 4 and 5, this significant difference is not observed. May I get explanation for this discrepancy between Figure 4 and table 4, 5? Also, can I get explanation why your model outperforms on PGD-$\ell_\infty$ unlike on BIM-$\ell_\infty$?

Fourth, in Figure 1, may I know the reason you compare tracked mean with mini-batch mean "before training"? I think a comparison with mini-batch mean at the "convergence" could be more meaningful because 1) the concern is the data distributional change caused when tracked mean is used 2) during the course of learning the mini-batch mean would change.

Fifth, in Madry et al (2016) https://arxiv.org/pdf/1706.06083.pdf, the reported robust accuracy of WideResnet on CIFAR10 is 45.8% agianst $\ell_\infty$ PGD attacks of $\epsilon=8/255$. In your table 4, I can see your networks have much better results. Do you think this is because of BatchNorm w/o tracking and RobustNorm? Or because of the network difference? Or any other things?

Thank you!

---

### Decision · Program_Chairs · 2019-12-19

**Decision:**

Reject

**Comment:**

 This paper presents an empirical analysis of the reasons behind BatchNorm vulnerability to adversarial inputs, based on the hypothesis that such vulnerability may be caused by using different statistics during the inference stage as compared to the training stage. While the paper is interesting and clearly written, reviewers point out insufficient empirical evaluation in order to make the claim more convincing.